# RNA Interference Therapeutics for Chronic Hepatitis B: Progress, Challenges, and Future Prospects

**DOI:** 10.3390/microorganisms12030599

**Published:** 2024-03-17

**Authors:** Laura Sneller, Christine Lin, Angie Price, Shyam Kottilil, Joel V. Chua

**Affiliations:** Division of Clinical Care and Research, Institute of Human Virology, University of Maryland School of Medicine, Baltimore, MD 21201, USA; lsneller@ihv.umaryland.edu (L.S.); christine.lin@ihv.umaryland.edu (C.L.); aprice@ihv.umaryland.edu (A.P.); skottilil@ihv.umaryland.edu (S.K.)

**Keywords:** hepatitis B virus, chronic hepatitis B, RNA interference, small interfering RNA, hepatitis B surface antigens, covalently closed circular DNA

## Abstract

Chronic hepatitis B (CHB) is a global health challenge that can result in significant liver-related morbidity and mortality. Despite a prophylactic vaccine being available, patients already living with CHB often must engage in lifelong therapy with nucleoside analogues. However, the potential of RNA interference (RNAi) therapeutics as a promising avenue for CHB treatment is being explored. RNAi, particularly using small interfering RNA (siRNA), targets viral RNA that can be used to inhibit hepatitis B virus (HBV) replication. Several candidates are currently being studied and have exhibited varying success in reducing hepatitis B surface antigen (HBsAg) levels, with some showing sustained HBsAg loss after cessation of therapy. The dynamic evolution of RNAi therapy presents a promising trajectory for the development of effective and sustained treatments for CHB. This review highlights recent findings on RNAi therapeutics, including modifications for stability, various delivery vectors, and specific candidates currently in development.

## 1. Introduction

Chronic hepatitis B (CHB) infection continues to be a global public health burden, with an estimated 296 million people afflicted worldwide [1]. Unmitigated, CHB can lead to liver cirrhosis and hepatocellular carcinoma (HCC) and accounts for more than 820,000 deaths per year [1]. Two classes of therapeutics are approved to treat CHB: nucleos(t)ide analogues (NA) and interferons (IFN). The current goal of treatment is to achieve functional cure, defined by sustained hepatitis B surface antigen (HBsAg) loss and hepatitis B virus (HBV) DNA suppression. Though nucleos(t)ide analogues suppress viral load and are associated with a significant reduction in progression to cirrhosis and HCC, rates of patients achieving functional cure are low [2], and most patients require lifelong NA therapy due to the endurance of covalently closed circular DNA (cccDNA) [3]. Therefore, new therapeutics are being explored to provide better alternatives for achieving a functional cure in patients with CHB.

Chronic hepatitis B is the result of a persistent hepatitis B virus (HBV) infection. HBV is a small DNA virus whose infectious virion (Dane particle) consists of a double-shelled sphere of a lipid envelope embedded with surface antigens (HBsAg). Inside the envelope is the nucleocapsid that holds encapsulated viral relaxed circular DNA (rcDNA) and polymerase protein. The virus attaches to the host cell membrane, and it is disassembled and transported intracellularly into the nucleus [4]. After the nucleocapsid travels to the nucleus of the host cell, the rcDNA is converted to cccDNA, which serves as a template for the transcription of pregenomic RNA (pgRNA) and other viral mRNAs. The cccDNA is supported by the HBx protein [4]. The HBV genome is remarkably compact and contains four viral open reading frames (ORFs) (Figure 1). The polymerase (P) ORF encodes the polymerase protein that replicates the HBV DNA via reverse transcription of the pregenomic RNA, which is then transcribed by the cellular RNA polymerase to form cccDNA [4]. The core (C) ORF encodes the viral nucleocapsid (HBcAg) and the hepatitis B e antigen (HBeAg) [4]. The ORF X encodes the HBx protein that aids in signal transduction, DNA repair, and inhibition of protein destruction [4]. The ORF S is responsible for expressing the HBsAg proteins and encodes for the three coterminal HBs proteins, with the Large-HBs corresponding with preS1, preS2, and S, the Middle-HBs with preS2 and S, and the Small-HBs with gene *S* [5]. The HBs proteins are released into the serum as HBsAg subviral particles and are essential for measuring the clinical phase of HBV infection in patients [6]. 

The major hepatitis B viral mRNAs are the C-mRNA (or pregenomic RNA), Pre-S mRNA, S-mRNA, and HBx-mRNA. The pregenomic RNA, 3.5 kb long, is the longest of the mRNA and is the template for both the reverse transcription of the relaxed circular DNA and for the translation of viral polymerase and core proteins [7]. The Pre-S mRNA is the primary template for the L-HBs protein (which contains Pre-S1, Pre-S2, and S proteins), whereas S-mRNA is the principal template for the M-HBs (contains Pre-S2 and S) and the S-HBs proteins [7]. The HBx-mRNA is the shortest (0.7 kb long) and is responsible for transcription of HBx protein, an important modulatory of HBV signal transduction. All HBV mRNAs end at the common polyadenylation signal located in the ORF C region, an important consideration for anti-HBV RNA interference (RNAi) therapies. HBV RNAi that targets the ORF X region can potentially interfere with all the mRNA transcripts, resulting in a broad reduction in all HBV products.

Among the new HBV treatments under development, RNA interference (RNAi) therapeutics have shown promise [9]. Small interfering RNA (siRNA) comprises two strands, a sense and an antisense strand. The antisense strand is complementary to the target sequence of mRNA and, thus, serves to guide the molecule. After being endocytosed, the siRNA interacts with the RNAse III endonuclease called Dicer, the RNAse Argonaute, and an RNA-binding cofactor to assemble the RNA-induced silencing complex loading complex (RLC). After the sense strand is discarded, a mature RNA-induced silencing complex (RISC) is formed. It can subsequently bind to the target mRNA with the complementary antisense strand and silences gene expression through mechanisms such as mRNA degradation [10]. RNAi acts to reduce the production of HBsAg by utilizing double-stranded RNA (dsRNA) to inhibit HBV messenger RNA (mRNA) translation [11]. Suppressing HBsAg to low levels is an important surrogate outcome that may predict HBsAg seroclearance and lead to functional cure [12]. This review summarizes the recent findings investigating RNAi therapeutics, including modifications to stabilize RNAi, various forms of vectors, and specific RNAi therapeutics that have been developed. This paper scrutinized studies based on efficacy, specificity, delivery systems, and safety profiles to assess the potential of RNAi as a therapeutic avenue for chronic HBV infection.

## 2. Hepatitis B RNA Interference

RNAi treatment holds promise for addressing conditions like chronic hepatitis B by effectively inhibiting the replication of the virus (Figure 2). This therapeutic approach targets crucial RNA molecules, including pgRNA and mRNA, to impede the formation of cccDNA. RNAi can be employed to hinder the translation of viral proteins crucial for cccDNA development (Figure 2). The compactness of the HBV genome restricts sequence plasticity, minimizing the virus’s ability to evade silencing sequences [13]. Moreover, the targeted nature of RNAi, focusing on the viral transcript, offers the potential to minimize undesired side effects and cellular toxicity [14,15]. HBV is differentiated into ten genotypes (A–J), with distinct geographical distribution [16,17]. RNAi mitigates the selection of escape mutants and different genotypes of HBV by concentrating on conserved regions of the HBV genome. The ability to introduce multiple siRNAs targeting diverse sequences further diminishes the likelihood of mutant development, catering to patients with a variety of circulating HBV genomes [18].

For siRNA antivirals to be effective, structural optimization is critical to prevent siRNA degradation in human serum [20]. Various chemical modifications can enhance siRNA efficacy, resulting in improved gene silencing at lower doses and reduced dosing frequency [21]. Incorporating a combination of deoxyribonucleotides, phosphorothioate linkages, and 2′-OH modifications has been identified as enhancing in vitro stability without compromising silencing efficacy [22]. Modified siRNAs have demonstrated superior performance in vivo, significantly inhibiting HBV viral replication in mice, with efficacy maintained for up to 6 weeks with single weekly doses [23]. Altritol, a six-membered ring replacing ribose in the siRNA structure, has been shown to improve HBV replication silencing and exhibit enhancements in immunostimulation, albeit with some observed hepatoxicity in vivo [24]. The inclusion of 2′-O-guanidinopropyl-modified ribonucleotides has demonstrated improved stability, immunostimulatory profiles, and silencing efficacy [25]. Additional modifications, such as asymmetric chemical modification of 3′-overhangs and incorporating a 5′-end triphosphate-modified siRNA, have shown increased silencing activity and long-term suppression [26,27]. Watson–Crick face modification of adenosine is another strategy for modulating immune system stimulation by siRNAs [28].

## 3. Delivery

### 3.1. Non-Viral Vectors

Delivery systems for siRNAs into target cells constitute crucial elements in designing an effective treatment strategy. Non-viral vectors, encompassing polymers, aptamers, peptides, liposomes, antibodies, and lipid nanoparticles, play a pivotal role [29]. Lipid nanoparticles (LNPs) with a core–shell structure, comprising cholesterol, phospholipids, and cationic ionizable lipids in the outer shell and cationic ionizable lipids, cholesterol, and RNA in the inner core, present a versatile delivery system [30]. LNPs exhibit advantageous ionizable behavior, enhancing endosomal escape, displaying self-assembly, and reducing toxicity [29]. For treating chronic hepatitis B, incorporating hepatocyte-targeting compounds such as galactopyranoside cholesterol and PEG can enhance stability in the blood. Ensuring uniform LNP size is essential to prevent sequestration during circulation, and modifications to avoid endosomal entrapment can improve their delivery efficiency [29].

In a study utilizing a nontoxic lipid-based vector system to deliver PEGylated siRNA to the liver showed that this approach significantly reduced HBV replication in vivo. Systemic administration of these nanoparticles to mice with HBV resulted in a notable suppression of HBV replication markers, achieving up to a threefold reduction over 28 days [31]. Guanidino-propyl-modified siRNAs in anionic polymer-containing lipoplexes demonstrated efficient delivery to the liver, effectively inhibiting viral replication [32].

Another lipid-based delivery system strategy is stable nucleic acid lipid particles (SNALPs). In a 2005 study evaluating the efficacy of siRNA delivered via SNALP to mice with HBV, this vector exhibited improved efficacy and a longer half-life in plasma and the liver, reducing HBV DNA levels in the serum for up to 6 weeks [33].

The conjugation of siRNAs to N-acetylgalactosamine (GalNAc/NAc) has gained prominence for liver-targeting therapy. Derivatives of GalNAc, with enhanced binding to asialoglycoprotein receptors on hepatocytes, enable liver-specific delivery of siRNAs [34]. Coinjecting a melittin-like peptide (MLP) targeted to hepatocytes, along with siRNA conjugated to cholesterol, led to a multiple log reduction in HBV gene expression and viral DNA titer with a prolonged duration of effect in mice models [35]. Modifications resulting in a triantennary GalNAc ligand facilitated dose-dependent and sustainable gene silencing for over nine months [36]. This triantennary GalNAc–siRNA conjugate was further stabilized through the modification of the 5′ siRNA [37].

An additional delivery strategy under study involves the nonviral episomal replicating vector pEPI-1, whose function relies on a transcription unit linked to a scaffold/matrix attachment region. Injecting HBV-replicating cells with pEPI-RNAi resulted in a remarkable reduction in HBV RNA expression for up to 8 months post-transfection [38]. A newly designed vector with an RNA polymerase II-driven gene cassette demonstrated efficient expression and processing of multiple miRNAs, achieving potent HBV silencing in cell culture models [39].

### 3.2. Viral Vectors

Viral vectors are pivotal in delivering therapeutic genes through cell transduction [40]. Among the extensively studied candidates for siRNA delivery to treat chronic hepatitis B are recombinant lentiviral vectors (LVs), adenoviral vectors, and adeno-associated viral vectors (AAVs) [41,42,43,44], all exhibiting efficient transduction to the liver.

Lentiviral vectors have demonstrated successful delivery of HBV-targeting RNAi activators to hepatocytes, inhibiting HBV replication markers without toxicity in neonatal HBV transgenic mice [45]. Moreover, LVs were utilized to introduce single-guide RNA (sgRNA) into human hepatoma PLC/PRF/5-CBE cells, reducing HBs mRNA levels and decreasing HBsAg secretion [36]. Concomitant injection of HBV plasmids with a lentivirus targeting HBV small hairpin RNA (shRNA) induced an RNAi response, reducing HBsAg in mouse serum and inhibiting HBeAg [46]. LVs were also employed to transduce sh-1580, an anti-HBx shRNA, effectively suppressing HBV replication and the presence of HBV cccDNA in cell culture systems [47]. However, in vivo, LVs exhibit low efficiency in transducing adult hepatocytes and may pose an oncogenic risk through chromosomal integration [13] as well as permanent integration of the viral genome into the human genome, requiring lifelong monitoring [48].

Recombinant and helper adenoviral vectors were studied for delivering anti-HBV siRNA to hepatocytes. Recombinant adenoviral vectors U6 shRNA 5 and U6 shRNA 6 significantly reduced HBV markers in transfected hepatocytes and a murine hydrodynamic injection model of HBV replication [44]. Delivery of siRNA treatment using a recombinant adenovirus suppressed HBV gene expression and replication for up to 26 days in HBV transgenic mice. Helper-dependent adenoviral vectors efficiently delivered shRNAs and artificial primary miRNAs to the mouse liver, inhibiting HBV replication [49]. However, this vector candidate may induce high innate immunostimulation, resulting in relatively short-term transgene expression [13].

Adeno-associated viral vectors are known for their safety and efficacy in human applications [33] and encompass approximately 100 serotypes, with AAV8 displaying liver tropism [29]. Despite initial concerns about potential oncogenic activity, further investigation alleviated these concerns [16]. Recombinant AAVs, including single-stranded AAVs (ssAAVs) and self-complementary AAVs (scAAVs), have demonstrated efficacy in suppressing HBV protein, mRNA, and replicative DNA in transgenic mice for up to 120 days [50]. scAAVs delivering anti-HBV sequences showed no toxicity or innate immunostimulation, significantly suppressing HBV replication markers for up to 32 weeks [42]. The AAV8 vector co-expressing HBV shRNA and TuD (or Tough Decoy RNAs, a bi-cistronic AAV vector co-expressing a shRNA with a second RNA hairpin) outperformed other strategies, exhibiting higher efficiency and persistent reduction of HBV [51]. Additionally, AAV8–amiRNA135, an artificial microRNA (amiRNA) recombined with an expression AAV vector, reduced HBsAg and HBeAg in HBV transgenic mice, maintaining low levels of these antigens for up to 15 months [52]. Sequential use of double strands (ds) AAV8 and dsAAV9 achieved a longer anti-HBV effect, demonstrating the efficacy of combining RNAi and multiple treatments with different AAV serotypes [53]. Despite their promising efficacy and safety profile, AAVs may induce immune responses and early clearance, which can be addressed by in silico reconstruction of hepatotropic capsid sequences to evade preexisting immunity [32,54].

## 4. Current RNAi Research

Several RNAi therapeutics for HBV are in various stages of development. Table 1 and Table 2 summarizes as well as compare and contrast select HBV RNAi agents which are in early clinical stages of development and that are further described in this section.

### 4.1. ARB-1467

ARB-1467 comprises three RNAi triggers that target the S and X ORFs and is administered intravenously via lipid nanoparticles. According to a study following a 12-week trial on patients being treated with NA and three monthly doses of ARB-1467, the mean reduction in HBsAg was from 0.6 to 0.9 log IU/mL and was sustained for 12 weeks [60]. A biweekly dosing regimen was tested and showed a greater reduction in HBsAg than monthly dosing, with a maximum individual HBsAg decrease of 2.7 log IU/mL. This reduction was sustainable at ten weeks [60]. The development of ARB-1467 was discontinued in March 2019 as the focus was shifted to the study of AB-729 [59]. 

### 4.2. ARC-520

ARC-520 is an intravenously administered siRNA conjugated to cholesterol with two triggers targeting overlapping regions in the X ORF of the HBV genome. It was well tolerated in patients with chronic hepatitis B in a phase 1 randomized, double-blind, placebo-controlled trial [58]. A substantial reduction in HBsAg levels in treatment-naïve HBeAg-positive patients was observed in a phase 2 trial, but this result was not seen in HBeAg-negative patients or those previously on NA treatment [57], which could be attributed to HBsAg expression from integrated copies of HBV that lacked the ARC-520 target site deleted during integration [57]. Furthermore, in two phase 2 trials, ARC-520 showed moderate decreases in HBsAg in both HBeAg-positive and negative NA-treated patients that was sustained for up to 85 days [56]. In a phase 1b study, eight patients received up to nine monthly doses of 4 mg/kg ARC-520 for eight to ten months after initial dosing. Of the HBeAg-negative patients, one achieved HBsAg seroclearance while the others had an average reduction in HBsAg of 0.4 log_10_ IU/mL. Of the HBeAg-positive patients, one achieved seroclearance, and the other two had HBsAg reductions of 1.7 log_10_ IU/mL and 3.5 log_10_ IU/mL [55]. The development of ARC-520 was terminated in 2016 due to the toxicity of the EX1 dynamic polyconjugate delivery vehicle in nonhuman primates [57].

### 4.3. ALG-125755

ALG-125755 is a subcutaneous GalNAc-conjugated siRNA that has a trigger targeting the S ORF. In mice, it was shown to preferentially reduce HBsAg by 1.5 log_10_ IU/mL, which was maintained for six weeks [75]. A three-part, double-blind, randomized, placebo-controlled phase 1a/1b study with virally suppressed chronic hepatitis B patients receiving either 50 mg or 120 mg doses of the siRNA found that dose-dependent reductions in HBsAg were observed over 90 days following a single dose of ALG-125755 [74]. The study is currently ongoing to evaluate the safety and efficacy of higher doses (e.g., 320 mg) [74].

### 4.4. RG-6346

RG-6346 is a subcutaneous GalNAc-conjugated siRNA with a trigger targeting the ORF S. In a phase 2a trial where patients were treated with both NA and four monthly doses of RG-6346, 92% achieved a greater than 1 log_10_ IU/mL decrease in HBsAg, and the suppression was maintained for up to 64 weeks [65]. It was associated with self-resolving ALT flares consistent with treatment-induced enhanced immune response, and the most common adverse events were related to mild injection site reactions [65]. 

### 4.5. AB-729

AB-729 is a GalNAc-conjugated siRNA with one trigger site targeting the HBV X protein and is administered subcutaneously. A phase 2 trial for NA-treated patients observed the effects of the siRNA at different doses and intervals, resulting in an HBsAg reduction from 1.86 to 2.16 log_10_ IU/mL, which was maintained for up to 48 weeks [64]. HBsAg decline was also associated with reductions in total HBV RNA, pgRNA, and large and middle HBsAg isoforms in both HBV DNA-positive and negative patients [63]. It was also found that ARB-729 caused increased HBV-specific immune responses and mild to moderate ALT elevations, suggesting immune reawakening in patients with chronic hepatitis B [62]. Another study, following the efficacy of AB-729 on patients with chronic hepatitis B, showed HBsAg declines ranging from −0.62 log_10_ to −2.14 log_10_ IU/mL that were sustained for 12 weeks after a single dose [61].

### 4.6. VIR-2218

ALN-HBV is a GalNAc-conjugated siRNA that targets highly conserved sequences in the ORF X. This RNAi therapeutic showed specific, potent, and durable silencing of HBV viral transcripts and HBsAg in chimpanzees [76]. Furthermore, in a trial observing the effects of this treatment on healthy volunteers, it was found that ALN-HBV caused post-treatment ALT elevations in 28% of participants [70], which may be due to off-target RNAi-mediated effects [66,67]. This molecule was modified to reduce off-target binding by substituting a glycol nucleic acid within the seed region [68], making VIR-2218. 

VIR-2218 is a GalNAc-conjugated siRNA with a trigger targeting the HBV ORF X and is administered subcutaneously. VIR-2218 showed favorable pharmacokinetics in healthy volunteers with no severe or serious adverse events and resulted in lower ALT levels than its parent molecule (ALN-HBV) [69]. A trial following NA-treated patients with two monthly doses of VIR-2218 for 48 weeks found that 70.8% of patients had a greater than 1 log_10_ IU/mL decline in HBsAg. Most participants achieved maximum mean HBsAg decline by week 16. But response was not sustained at 48 weeks for those receiving 20 and 50 mg doses of VIR-2218. Slightly more than a third (36.4%) of participants who received either a 100 mg or 200 mg dose exhibited sustained HBsAg reduction [70]. In another study, VIR-2218 administration was associated with dose-dependent HBsAg reduction, of which the mean reduction was greatest at week 20 in participants receiving 200 mg: 1.65 log_10_ IU/mL, with an HBsAg reduction maintained at 0.87 log_10_ IU/mL at week 48 [70].

### 4.7. JNJ-3989

JNJ-3989 is a GalNAc-conjugated siRNA with two triggers for the ORFs X and S in the HBV genome and is administered subcutaneously. In a phase 2a trial, this siRNA was combined with NA therapy with/without the capsid assembly modulator (CpAM), JNJ-6379, to treatment-naïve and NA-suppressed CHB participants [73]. This combination therapy was well tolerated and resulted in HBsAg reductions of greater than or equal to 1 log_10_ IU/mL from baseline for 97.5% of participants at the nadir and decreases in HBsAg were similar in both HBeAg-positive and HBeAg-negative participants [73]. These reductions persisted in 38% of patients up to 36 days after their last dose [73]. Furthermore, the triple combination of RNAi (JNJ03989), CpAM (JNJ-6379), and NA led to a 1.7 log reduction in HBsAg on day 113 in 12 CHB participants, as well as the suppression of other viral products (HBV DNA, HBV RNA, and HBsAg) [71]. Another study showed that the administration of JNJ-3989 to NA-experienced or naïve patients could bring about 1.3–3.8 log_10_ IU/mL HBsAg reductions [77]. Finally, a study with 470 patients with chronic hepatitis B found that JNJ-3989 had a dose-dependent response. They observed that this treatment rarely led to HBsAg seroclearance, despite patients having clinically meaningful reductions in HBsAg [72]. 

## 5. Combination Therapy

Combination therapies that include RNAi and licensed medications have been studied as potential means to target different parts of the HBV lifecycle. The combination of siRNA JNJ-3989 with the capsid assembly modulator JNJ-6379 was tested in NA-treated patients, resulting in a 1.8 log_10_ IU/mL decrease in HBsAg in those receiving both treatments, whereas the JNJ-6379 monotherapy only led to a 0.07 log_10_ IU/mL decrease [73]. The most common adverse event reported in this trial was flu-like symptoms [73]. VIR-2218 was also tested with PEGylated interferon alpha-2a in NA-treated CHB patients, and a HBsAg reduction of 2.03 log_10_ IU/mL was observed for those receiving the combination treatments. Those in the monotherapy group receiving only VIR-2218 experienced a HBsAg decrease of 1.89 log_10_ IU/mL [78]. There are currently more combination treatment trials being evaluated, such as the triple combination of RG6346 with the CpAM RO7049389 and NA, RG6346 with a TLR7 agonist and NA, RNAi with PEG-IFN and NA, or CpAM with TLR agonist, and NA [79]. AB-423 was also found to have synergistic antiviral activity in combination with NAs in vitro. When given with entecavir or ARB-1467 in HBV-infected liver-humanized mice, it resulted in higher antiviral activities, consistent with in vitro findings [80].

Synergistic effects of shRNA and siRNA were observed with lamivudine, and results showed a better anti-HBV effect than either monotherapy in transfected cell cultures [81,82]. Furthermore, the combination of siRNAs targeting different sites in the HBV genome showed effective inhibition of viral replication and antigen expression, compared with the treatment of only one siRNA [83]. This was also shown in the combination of siHBV-74 (targeting the X ORF) with siHBV-77 (targeting the S ORF) in a pHBV mouse model [84]. Another study focusing on the effects of siRNA administration and vaccine delivery to mice found that using siRNA to target the programmed death protein’s ligand increased the efficacy of the heterologous prime–boost therapeutic vaccination scheme TherVacB [85].

Another recent development in the world of gene therapy for CHB is antisense oligonucleotide (ASO) treatment; combinations of siRNAs with ASO therapies have also been studied. A promising candidate of this class of molecules is Bepirovirsen (GSK3228836), a subcutaneously-administered Gal-NAc-conjugated 2′-O-methoxyethyl gapmer targeting the ORF X. Phase 2 clinical trials with this ASO have yielded encouraging reductions in HBsAg [86]. It was found that the treatment of HepG2.2.15 cells and HBV-infected mice with the siRNA ALG-125903 and ASO ALG-020579 as well as other anti-HBV agents like NAs and capsid assembly modulators demonstrated significant synergy and increased HBsAg suppression [87].

## 6. Conclusions

In recent times, substantial progress has been made in RNA silencing and interference, positioning it as a viable therapeutic avenue for various chronic diseases. Notably, synthetic siRNAs have demonstrated efficacy in silencing HBV replication in vitro and in vivo. Trials addressing the treatment of chronic hepatitis B with diverse RNAi modalities (Table 1) have consistently yielded positive outcomes, manifesting in robust and sustained reductions in HBsAg levels and, in select cases, instances of HBsAg seroclearance (Table 2). These encouraging findings present an optimistic perspective on the potential realization of a functional cure for chronic HBV infection.

Despite the promising results from diverse RNAi studies targeting HBV, the feasibility of achieving enduring HBsAg seroclearance remains uncertain. Addressing this uncertainty necessitates protracted trial durations with extended follow-up intervals dedicated to monitoring HBsAg seroclearance. Additionally, further investigations are imperative to elucidate the impact of RNAi treatment on distinct chronic hepatitis B subgroups, encompassing both HBeAg-positive and negative patient cohorts. Challenges such as potential off-target effects, immunostimulation, enhanced target site selection, siRNA design considerations [14], and meticulous dose regulation [88] pose substantial hurdles in developing RNAi-based treatments.

Continued research efforts are essential to comprehensively establish the biological activities of HBV transcription, viral cccDNA, and the intricacies of RNAi mechanisms. Furthermore, exploring the efficacy and safety profiles of combination therapies, integrating RNAi alongside other potential gene therapies, and established nucleos(t)ide analogues merits dedicated attention. The dynamic evolution of RNAi therapy instills confidence in the prospective effectiveness of treatments for chronic hepatitis B, setting an optimistic trajectory for the future of therapeutic interventions.

## Figures and Tables

**Figure 1 microorganisms-12-00599-f001:**
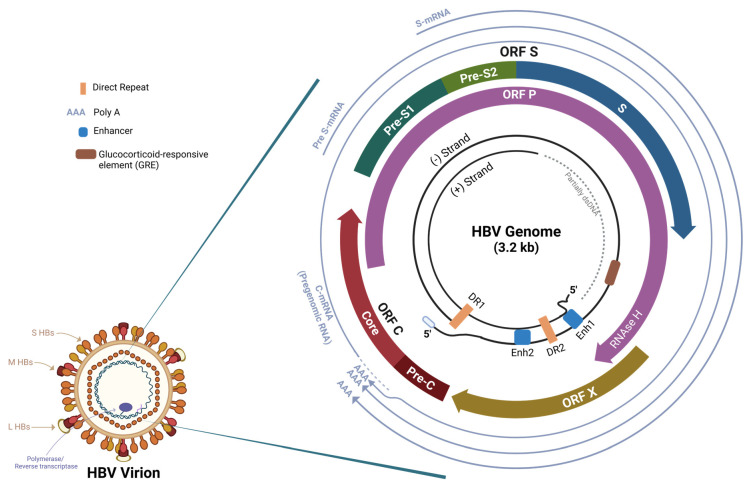
Comprehensive schematic of the HBV genome. An overview delineating key genomic regions and regulatory elements in the viral genome (dsDNA, double-stranded DNA; ORF, open reading frame; DR, direct repeat; Enh, enhancer; RNAse H, ribonuclease H). The four major hepatitis B viral messenger RNAs (mRNA), the C-mRNA (or pregenomic RNA), Pre-S mRNA, S-mRNA, and the HBx-mRNA (not shown in illustration), all end at the common polyadenylation (Poly A) signal located in the core open reading frame (ORF C). The envelope of the HBV virion contains the small (S-HBs), middle (M-HBs), and large (L-HBs) hepatitis B surface proteins, the majority existing in dimers, and encases an icosahedral nucleocapsid consisting of core proteins that in turn encapsulates a double-stranded circular genome and a polymerase/reverse transcriptase. The HBV genome consists of a complete circular negative strand and an incomplete positive strand (see innermost circles inside the genome). The pregenomic RNA is the template for both the reverse transcription of the relaxed circular DNA and for the translation of viral polymerase and core proteins [7]. The Pre-S mRNA is the primary template for the L-HBs protein (which contains Pre-S1, Pre-S2, and S proteins), whereas S-mRNA is the principal template for the M-HBs (contains Pre-S2 and S) and the S-HBs proteins. HBx-mRNA is transcribed from the beginning of the ORF X region and ends at the Poly-A region. Figure adapted from Liang [4] and Seeger [8]. Illustration created by authors using BioRender.com on 12 March 2024.

**Figure 2 microorganisms-12-00599-f002:**
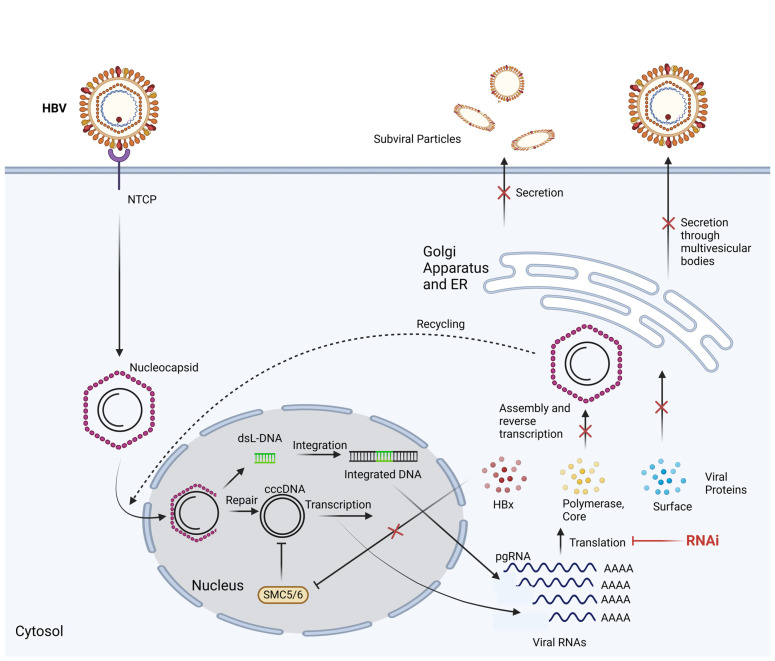
Mechanism of RNA interference in the disruption of the HBV replication cycle. Through the utilization of RNA interference, the translation of viral RNAs can be inhibited, and the prevention of HBV replication can be achieved (sodium taurocholate co-transporting polypeptide, NTCP; double-stranded linear DNA, dslDNA; structural maintenance of chromosome complex 5/6, SMC5/6; pregenomic RNA, pgRNA). Figure adapted from Maepa et al. [19]. Illustration created by authors using BioRender.com, accessed on 5 March 2024.

**Table 1 microorganisms-12-00599-t001:** Selected RNAi trials for treatment of chronic hepatitis B.

Drug	Number of Triggers	siRNA Target	Delivery	Phase	Status	References
ARC-520	2	ORF X	Cholesterol	2	Completed/Terminated	[55,56,57,58]
ARB-1467	3	ORF S and X	LNP	2a	Completed	[59,60]
AB-729	1	ORF X	GalNAc	2	Completed	[61,62,63,64]
RG-6346	1	ORF S	GalNAc	2a	Completed	[65]
VIR-2218	1	ORF X	GalNAc	1/2	Completed	[66,67,68,69,70]
JNJ-3839	2	ORF S and X	GalNAc	1/2a	Completed/Terminated	[71,72,73]
ALG-125755	1	ORF S	GalNAc	1a/1b	Completed	[74]

**Table 2 microorganisms-12-00599-t002:** A Comparison of RNAi in advanced stages of clinical development and their trial results.

Drug	Phase of Development	Total Number of Participants Receiving Treatment	Maximum Reduction in HBsAg (log_10_)	Number of Days Suppression Was Maintained	References
ARC-520	2	58	1.4	>85 days	[56]
ARB-1467	2a	12	2.7	70 days	[60]
AB-729	2	34	2.16	336 days	[64]
RG-6346	2a	16	1.91	448 days	[65]
VIR-2218	2	24	1.65	336 days	[70]
JNJ-3839	2a	40	>2.5	113 days	[71]

## Data Availability

No new data were created or analyzed in this study. Data sharing is not applicable to this article.

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
