# Peer review of "RNA Interference Therapeutics for Chronic Hepatitis B: Progress, Challenges, and Future Prospects"

_microorganisms, 2024, doi:10.3390/microorganisms12030599_

Round 1

Reviewer 1 Report

Comments and Suggestions for Authors

General comment

The topic of RNAi and siRNA as therapeutic concept against CHB is highly relevant and interesting. However, while reading the manuscript, I realized that is was very immature and partly disorganized. Thus, I stopped careful reading at page 5. I mention below those points which I have noted for correction, if a revision is submitted.

Specific points

1.      Figure 1. The figure contains many mistakes and is not suitable for the aims of the review. The authors should discard this failed attempt to adapt a map of the HBV genome. Unfortunately, they selected a grossly wrong map of the HBV DNA shown in ref. 83.

a.      The primary structure of the HBV DNA is secondary for this paper.

b.      Much more relevant would be the map of the HBV mRNAs in relation to the HBV ORFs.

c.      However, the ORFs should be correctly shown. ORFs have a direction from 5’ to 3’ which is usually shown by a short symbol for an arrow.

d.      The 3 HBV surface proteins are all encoded by one ORF with two internal start codons. PreS1 , preS2 and S are regions of the S ORF.

e.      The direction of the Pol ORF is drawn wrong. The RNase H domain is the carboxy-end.

f.       ORF X is misplaced

g.      The preC sequence does not overlap with the Pol ORF.

2.      Fig. 2 is overall acceptable.

a.      However, budding of HBV or HBsAg particle from the cell surface does not occur. HBV particles and HBsAg particles are released via different pathways; the virus via multivesicular bodies, the subviral particles via the Golgi.

3.      L32. References 85-100 are not listed. The references end with nr. 84.

a.      Furthermore, the sentence in L32 is partially wrong. Integration of HBV DNA does not require permanent NA therapy.

4.      L34-45. This introductory text should refer to figures 1 and 2. Why not? However: see points 1 and 2.  

Comments on the Quality of English Language

The English is ok. But references are partly missing.

Author Response

Thank you for your time in reviewing our manuscript. We highly appreciate your insightful comments, and we think this has greatly improved the clarity and coherence of our review paper. We have remedied the citation numbers so that they correspond correctly to the bibliography and removed the sentence saying that the integration of HBV DNA requires permanent NA therapy. We also modified Figure 1 with the feedback you provided and have used a different genome diagram as a basis for our figure in hopes that it is more accurate than the previous one. More specifically, we included arrows so that the directions of the ORFs are clearer and the PreS1, preS2, and S regions in the S ORF. Additionally, we corrected the placement of the RNase H domain, the pre-C sequence, and the X ORF. In addition, we have specified in Figure 2 that the virus is released via multivesicular bodies as well as the subviral particles through the Golgi. Furthermore, we referred to both figures in the introductory paragraphs. Once again, thank you for your time and expertise and we would be grateful for any further suggestions that you may have.

Reviewer 2 Report

Comments and Suggestions for Authors

The main question the authors of the manuscript discuss is the potential of RNA interference (RNAi) therapy as a promising treatment for chronic hepatitis B. The review presents data from studies using small interfering RNA (siRNA) that can be used to inhibit hepatitis B virus (HBV) replication. The authors report current candidates for such therapy and their reduction potential in terms of lowering HBsAg levels. The authors also present the latest findings on RNAi therapy, stability modifications, and different delivery vectors.

The manuscript is original because it contains the latest information on the issue of chronic hepatitis B treatment, available treatment gaps and possible potential future therapies.  The conclusions drawn by the authors are consistent with the presented data from the studies. The authors outline questions for clarification in future studies and opportunities to combine therapies to achieve better viral clearance.

The references are relevant to the review data and the conclusions.

I have not recommendations for the tables and figures presented in the manuscript.

I consider that the authors should describe in their manuscript what criteria they used to search for and include all possible current studies on the topic of chronic hepatitis B treatment with RNAi. That way there will be no doubt that there are studies that may have been missed.

Author Response

Thank you so much for your time and effort reviewing our manuscript. We have added a sentence in the introductory text detailing our criteria for our literature search and have ensured that the studies included on RNAi therapy for HBV are up to date. We appreciate your time and expertise and would be grateful for any further suggestions that you may have for improving our review.

Reviewer 3 Report

Comments and Suggestions for Authors

Chronic hepatitis B virus (HBV) infection poses a significant global public health threat, contributing to substantial morbidity and mortality. Nucleotide or nucleoside analogs have significantly improved the quality of life for patients with chronic hepatitis B (CHB). However, achieving a functional cure remains challenging due to the persistent presence of covalently closed circular DNA (cccDNA) in infected hepatocytes. Numerous studies related to small interfering RNA (siRNA) and its therapeutic trials have been reported, demonstrating their effectiveness in inhibiting the expression of viral proteins. In their review article, Sneller et al. provide a comprehensive overview of the historical progression up to the present and discuss the effects of siRNAs, as well as delivery methods. This review offers invaluable insights for researchers and clinicians engaged in the field of HBV. While the paper is well-written and suitable for publication, several concerns have been raised and need to be addressed.

1.       In Fig. 1, the positional relationship of key genomic regions and regulatory elements such as aEnh1, Enh2, DR1, DR2, and ORF appears to be incorrect and requires modification.

2.       In Fig. 2, a pathway seems to indicate the generation of rcDNA to dslDNA within the nucleus. To clarify that this process originates from particles containing dslDNA, arrows should be added from the decapsidated figure on the left.

Author Response

Thank you for your time in reading and editing our manuscript. We have modified both figures in accordance with your suggestions. More specifically, for Figure 1, we corrected the positions of the key genomic regions and regulatory elements, and for Figure 2, we clarified the generation of dslDNA by adding arrows from the decapsidated figure. Thank you again for your time and expertise and we would be grateful for any further suggestions that you may have for improving our review.

Round 2

Reviewer 1 Report

Comments and Suggestions for Authors

General comment

The authors have rapidly resubmitted an initially incomplete but now improved manuscript. The current version covers quite well the principles of RNAi therapeutic concepts and preclinical or clinical trials. However, the introductory part on the HBV mRNAs and their function is still unsatisfactory and contains many inaccurate statements (points 1-7). Most important is to provide a map of the HBV mRNA (point 4a).

Further points to be corrected are listed below (10 and 14 are major).

Specific points

1.      L15. Explain here the abbreviation HBsAg.

2.      L37. Before the travel of the nucleocapsid, the virus entry and the release from the envelope must happen which should be briefly mentioned.

3.      L40. The stability and transcription of the cccDNA is supported, not maintained, by the viral HBx protein.

4.      L41. Fig. 1 is now improved.

a.      However, most relevant is the map of the HBV mRNAs in relation to the HBV ORFs in a paper on RNAi therapies. An optimal source is shown in fig. 1 from: C. Seeger, W.S.Mason/Virology479-480(2015)672–686

b.      The ORF S covers the entire open reading frame from the start of preS1 to the end of the gene S encoding the small HBs protein. The label “ORF S” should remain outside the circle. PreS1, preS2 and gene S should be written inside the colored circle. The preS2 region is 55 aa long, the pres1 region has 108 aa, S has 226 aa. This should be drawn to scale.

c.      The virus model at the left is unsatisfactory and may be replaced by a better one.

5.      L42. The “repair” of the HBV DNA after entry is not done by the HBV polymerase but by cellular factors. The HBV pol replicates the HBV DNA via reverse transcription of the pregenomic RNA which is again transcribed by the cellular RNA pol.

6.      L45. The sentence on the S ORF is unsatisfactory. It should explain what is shown in figure 1. The ORF S encodes the three coterminal HBs proteins, L-HBs with preS1, preS2 and S, M-HBs with preS2 and S, the small S-HBs protein encoded by gene S. This is also relevant for L229.

7.      L46. This sentence is also unsatisfactory. The HBs proteins are released to the serum as HBsAg subviral particles and are essential for measuring the clinical phase of HBV infection in patients.

8.      L148. What is sgRNA? What kind of cells are PLCE/PRF/5-CBE cells?

9.      L171. What is TuD?

10.   L182 and later. All HBV mRNAs are coterminal. Thus, a trigger targeting the X ORF targets to all HBV mRNAs. Whether the truncated mRNAs for core, pol or S proteins are still efficiently translated is not discussed. This should be done at least at L225-230 and L243-253 where repeatedly  HBx targeting siRNAs reduce HBsAg expression.

11.   L201,202. There is two times: “one achieved seroclearance”. Is this the same patient or two?

12.   L209. Is the 1.5 log reduction in mice or patients?

13.   L289, 302. What are HHBV-infected mice or HBV-infected mice? Were these mice “humanized “ in the liver?

14.   References. Many references are incomplete or just abstract of meetings, e.g., ref. 11, 14, 23, 24 etc. Citing abstracts may be useful, but it should be clearly recognizable.

Comments on the Quality of English Language

The language is not the problem, the terminology of HBV is partly suboptimal.

Author Response

The authors have rapidly resubmitted an initially incomplete but now improved manuscript. The current version covers quite well the principles of RNAi therapeutic concepts and preclinical or clinical trials. However, the introductory part on the HBV mRNAs and their function is still unsatisfactory and contains many inaccurate statements (points 1-7). Most important is to provide a map of the HBV mRNA (point 4a).

Further points to be corrected are listed below (10 and 14 are major).

REPLY SUMMARY: We would like to thank you sincerely for your thorough review and insightful comments which has been invaluable in improving our manuscript. Your expertise and attention to detail have significantly enhanced the quality and clarity of the manuscript. In terms of changes to the text, we have clarified the abbreviations and processes in Lines (line location may have changed due to edits) 15, 37, 40, 42, 45, 46, 148, 171, 201, 202, 209, 289, and 302 to improve the accuracy of the information given. We also edited Figure 1 to include the map of HBV mRNAs, the specific genes in the S ORF, and deliver a more satisfactory virus model. Furthermore, we discussed the HBV mRNAs’ coterminal nature and how siRNA triggers affect the translation of the different HBV proteins. Lastly, we edited the references to improve the clarity of what sources were being cited. Your constructive feedback and meticulous editing have undoubtedly strengthened the final publication, and we are deeply appreciative of your dedication to advancing the scholarly discourse in our field.

Point-by-point replies are also included below.

Specific points

  1. L15. Explain here the abbreviation HBsAg.

REPLY: Thank you for pointing this out. We have spelled out HBsAg in the abstract.

  1. L37. Before the travel of the nucleocapsid, the virus entry and the release from the envelope must happen, which should be briefly mentioned.

REPLY: We have added a sentence to address this: “The virus attaches to the host cell membrane, and it is disassembled and transported intracellularly into the nucleus”.

  1. L40. The stability and transcription of the cccDNA is supported, not maintained, by the viral HBx protein.

REPLY: To address this, we have revised this line to “The cccDNA is supported by the HBx protein.”

  1. L41. Fig. 1 is now improved.
  2. However, most relevant is the map of the HBV mRNAs in relation to the HBV ORFs in a paper on RNAi therapies. An optimal source is shown in fig. 1 from: C. Seeger, W.S.Mason/Virology479-480(2015)672–686.

REPLY: Thank you for this recommendation. We have further improved our Figure 1 by addition of major hepatitis B mRNAs particularly C-mRNA (pregenomic RNA), pre-S mRNA, and S-mRNA. All these major mRNAs terminates at the same poly-AAA region. In addition, we have expanded the description of the Figure 1 to explain further significance of the mRNAs.

  1. The ORF S covers the entire open reading frame from the start of preS1 to the end of the gene S encoding the small HBs protein. The label “ORF S” should remain outside the circle. PreS1, preS2 and gene S should be written inside the colored circle. The preS2 region is 55 aa long, the pres1 region has 108 aa, S has 226 aa. This should be drawn to scale.

REPLY: Thank you for this observation. We have adjusted Pre-S1, Pre-S2, and S regions to appropriate scale based on aa length. We have also adjusted location of ORF S label. See new and improved version of Figure 1.

  1. The virus model at the left is unsatisfactory and may be replaced by a better one.

REPLY: HBV virion has been modified and improved to illustrate better the L/M/S HBs proteins on the cell surface with addition of labels.

  1. L42. The “repair” of the HBV DNA after entry is not done by the HBV polymerase but by cellular factors. The HBV pol replicates the HBV DNA via reverse transcription of the pregenomic RNA which is again transcribed by the cellular RNA pol.

REPLY: This is a very helpful comment. We have modified this statement to “The polymerase (P) ORF encodes the polymerase protein that replicates the HBV DNA via reverse transcription of the pregenomic RNA, which is hen transcribed by the cellular RNA polymerase to form cccDNA”.

  1. L45. The sentence on the S ORF is unsatisfactory. It should explain what is shown in figure 1. The ORF S encodes the three coterminal HBs proteins, L-HBs with preS1, preS2 and S, M-HBs with preS2 and S, the small S-HBs protein encoded by gene S. This is also relevant for L229.

REPLY: The sentence on S ORF has been modified to: “The S ORF is responsible for expressing the HBsAg proteins and encodes for the three coterminal HBs proteins, with the Large-HBs corresponding with preS1, preS2, and S, the Middle-HBs with preS2 and S, and the Small-HBs with gene S.”

  1. L46. This sentence is also unsatisfactory. The HBs proteins are released to the serum as HBsAg subviral particles and are essential for measuring the clinical phase of HBV infection in patients.

REPLY: This sentence has been changed to “The HBs proteins are released into the serum as HBsAg subviral particles and are essential for measuring the clinical phase of HBV infection in patients.”

  1. L148. What is sgRNA? What kind of cells are PLCE/PRF/5-CBE cells?

REPLY: sgRNA (single-guide RNA) has been defined. PLCE/PRF/5-CBE cells are human hepatoma cells, and this has been added to the sentence.

  1. L171. What is TuD?

REPLY: TuD or Tough Decoy RNAs are bi-cistronic AAV vector co-expressing a shRNA with a second RNA hairpin and this has been defined in the sentence containing this word.

  1. L182 and later. All HBV mRNAs are coterminal. Thus, a trigger targeting the X ORF targets to all HBV mRNAs. Whether the truncated mRNAs for core, pol or S proteins are still efficiently translated is not discussed. This should be done at least at L225-230 and L243-253 where repeatedly  HBx targeting siRNAs reduce HBsAg expression.

REPLY: Thank you for this recommendation. This is a very important point, and we have added a paragraph that address the importance of targeting ORF X region as in reference to mRNA transcripts all ending at the poly-AAA region found downstream.

  1. L201,202. There is two times: “one achieved seroclearance”. Is this the same patient or two?

REPLY: We have modified this sentence for improved clarity. It now states: “Of the HBeAg negative patients, one achieved HBsAg seroclearance while the others had an average reduction of HBsAg of 0.4 log10 IU/mL. Of the HBeAg-positive patients, one achieved seroclearance, and the other two had HBsAg reductions of 1.7 log10 IU/mL and 3.5 log10 IU/mL, respectively.”

  1. L209. Is the 1.5 log reduction in mice or patients?

REPLY: Reduction is in mice. We have clarified this in our manuscript.

  1. L289, 302. What are HHBV-infected mice or HBV-infected mice? Were these mice “humanized “ in the liver?

REPLY: We have clarified this in the manuscript. These are HBV infected liver-humanized mice.

  1. References. Many references are incomplete or just abstract of meetings, e.g., ref. 11, 14, 23, 24 etc. Citing abstracts may be useful, but it should be clearly recognizable.

REPLY: We have edited the references mentioned to improve the clarity of the sources being cited.